# Decoding the lncRNA World: Comprehensive Approaches to lncRNA Structure and Interactome Studies

**DOI:** 10.3390/cells15020105

**Published:** 2026-01-07

**Authors:** Mihyun Oh, Bo Lim Lee, Srinivas Somarowthu

**Affiliations:** 1Graduate Program in Molecular and Cell Biology and Genetics, Graduate School of Biomedical Sciences and Professional Studies, College of Medicine, Drexel University, Philadelphia, PA 19102, USA; moh@nysbc.org; 2Department of Biochemistry and Molecular Biology, College of Medicine, Drexel University, Philadelphia, PA 19102, USA

**Keywords:** lncRNAs, RNA structure, chemical probing

## Abstract

Recent advances in sequencing technologies have highlighted long non-coding RNAs (lncRNAs) as key regulators that perform essential biological functions without encoding proteins. Despite growing interest, the molecular mechanisms of most lncRNAs remain poorly understood, with only a few characterized in detail. A promising strategy to elucidate these mechanisms is to explore their structure–function relationships. Such studies require advanced biophysical and biochemical methods due to the large size and structural complexity of lncRNAs. Equally important is the analysis of lncRNA interactomes, which reveal how lncRNAs engage RNA-binding proteins and other biomolecules to drive conformational and functional changes underlying diverse biological pathways. Ultimately, integrative approaches combining structural and interactome analyses will yield deeper insight into lncRNA function and uncover new therapeutic opportunities. This review highlights recent advances in elucidating lncRNA structure–function relationships by integrating biophysical, biochemical, and sequencing-based approaches to overcome challenges of size and heterogeneity, identify functional binding partners, and inform therapeutic target development.

## 1. Introduction

RNAs that play crucial biological roles yet do not encode proteins—such as XIST, H19, and RNase P—have been empirically recognized for decades. More recently, advances in high-throughput RNA sequencing have revealed that protein-coding genes comprise less than 2% of the human genome, even though much of the genome is transcribed into RNA in a highly regulated manner [1,2]. This discovery has driven interest in the vast repertoire of non-coding transcripts, particularly long non-coding RNAs (lncRNAs) [1], which are RNA molecules longer than 200 nucleotides lacking protein-coding potential.

Recent studies have revealed that lncRNAs play key roles in cell development, aging, and disease progression [3,4,5]. As such, lncRNAs are garnering significant interest as potential therapeutic targets against various diseases, including cancer and neurological disorders [6,7,8]. Despite their functional significance and therapeutic potential, lncRNAs’ mechanisms of action remain unknown. Do lncRNAs function through interacting with protein-binding partners, or do they function on their own? Until now, only a few lncRNAs, such as Xist, have been intensively studied, even though the number of lncRNAs in the human genome is estimated to be approximately 20,000 [9]. Notably, many lncRNAs have been implicated in human disease, including cancer and developmental disorders, yet their mechanisms of action remain poorly. This disparity underscores a major gap in our understanding of how lncRNAs function at the molecular level. Therefore, investigating the molecular mechanisms of lncRNAs is urgently needed.

This review highlights recent advances in elucidating lncRNA structure–function relationships through integrated biophysical, biochemical, and sequencing-based approaches. To build a conceptual foundation, we first provide a brief background covering lncRNA definitions, major mechanistic principles, disease relevance, and therapeutic implications. These topics provide the context for structural studies of lncRNAs and the challenges posed by lncRNA size, heterogeneity, and functional complexity. We next review experimental methods, discuss structure-based therapeutic strategies, and examine structure-guided analyses of lncRNA interactomes.

## 2. Long Non-Coding RNA (lncRNA)

### 2.1. Definition

Long non-coding RNAs (lncRNAs) have long been defined as RNA transcripts longer than 200 nucleotides (nt) without protein-coding potential, a threshold that remains the most used in the field. This cutoff, while practical for distinguishing RNA types during purification, is somewhat arbitrary [10,11]. In an effort to provide a more biologically relevant classification, Mattick et al. (2023) proposed dividing non-coding RNAs into three categories: small RNAs (less than 50 nt), RNA polymerase III (Pol III) and Pol V transcripts (50–500 nt), and lncRNAs (greater than 500 nt) [10,11]. However, given the substantial diversity of lncRNAs in terms of function, processing, and genomic origin, a single, universally applicable definition remains elusive. As a result, some researchers suggest narrowing the focus to lncRNAs transcribed from RNA polymerase II (Pol II) primary transcription units [11]. Taken together, while the ~200 nt threshold remains the standard for defining lncRNAs, efforts are ongoing to refine the definition to reflect biological function better and gain broader consensus in the field. For a comprehensive review of ncRNA definition, see Mattick et al. (2023) [11].

### 2.2. Expression and Conservation of lncRNA

LncRNAs possess many mRNA-like characteristics, in addition to being transcribed by RNA Pol II [12,13]. Many lncRNAs undergo extensive processing, including splicing, 5′ capping, and 3′ polyadenylation [2,13]. However, several features set lncRNAs apart from mRNAs. Their expression is often tightly restricted to specific tissues, developmental stages, and subcellular compartments, suggesting that lncRNAs play context-dependent functional roles [14,15]. In addition, lncRNAs are typically expressed at lower copy numbers, consistent with regulatory rather than housekeeping functions [15].

Compared to mRNAs, lncRNAs exhibit reduced sequence conservation. However, this does not imply a lack of functional conservation. Instead, specific features such as exon–intron architecture, splice junctions, and promoter regions show measurable conservation, although generally weaker than that observed for mRNAs [13,16]. Thus, it has been proposed that the identification of conserved sequence motifs can enable the exploration of the extensive noncoding genome to discover functional lncRNAs and prioritize key sequences for experimental validation within the already-identified lncRNAs [17].

## 3. Mechanisms and Functions of lncRNA

LncRNAs are increasingly recognized for their functional importance, with growing evidence highlighting their involvement in key biological processes such as development, aging, and disease progression [18]. Central to these roles is their ability to regulate gene expression through diverse and complex mechanisms. Depending on their subcellular localization, either in the nucleus or cytoplasm, lncRNAs employ distinct mechanisms to exert their function [19]. Nuclear lncRNAs regulate gene expression at the epigenetic and transcriptional levels by influencing processes such as histone modification, DNA methylation, and chromatin remodeling through interacting with various proteins and transcription factors [19]. For example, HOTAIR interacts with the PRC2 chromatin-remodeling complex to mediate gene repression, whereas HOTTIP associates with the WDR5/MLL complex to promote gene activation [20,21]. Additionally, lncRNA APOLO interacts with DNA to form R-loop structures, which help control the expression of specific genes [22]. More recently, lncRNA SLERT has been shown to regulate transcription by RNA polymerase I by altering the biophysical properties of the fibrillar center and dense fibrillar component in the nucleus [23].

Conversely, cytoplasmic lncRNAs regulate gene expression mainly at the post-transcriptional level. Through interactions with microRNAs (miRNAs) and various proteins, they impact RNA stability, translation, and RNA metabolism, thereby fine-tuning protein production and contributing to gene silencing pathways [19]. lncRNA PNUTS serves as a miR-205 sponge, increasing ZEB mRNA levels [24]. lncRNA TINCR binds differentiation mRNAs with STAU1 to stabilize them, ensuring normal epidermal differentiation [25].

Although additional research is required to fully elucidate the functions of mitochondrial long non-coding RNAS (mtlncRNAs), emerging evidence suggests that they are key regulators of mitochondrial gene expression, metabolism, and even nuclear gene expression [26]. For example, mtlncRNA lncMtDloop—named after the region of the mitochondrial genome that encodes the lncRNA—increases mitochondrial regulation and suppresses Alzheimer’s progression [27].

The ability to interact with proteins, DNA, and RNA enables lncRNAs to regulate gene expression [3,4,5]. Four archetypes of molecular functions that lncRNAs perform through interactions with other biomolecules have been proposed: signals, decoys, guides, and scaffolds (Figure 1) [28]. As signals, they act as rapid responders that convey the activity of transcription factors and signaling pathways without requiring protein translation [28]. For instance, COOLAIR is induced by cold and marks temperature-responsive gene expression in plants [29]. As decoys, lncRNAs, such as GAS5, bind and sequester regulatory proteins, including transcription factors, thereby modulating pathways like steroid hormone signaling [28,30]. Other lncRNAs function as molecular sponges for microRNAs (miRNAs). For example, PCAT1 acts as a molecular sponge for miR-145-5p, while UCA1 serves as a molecular sponge for miR-204, miR-143, and miR-331-3p [31,32,33,34]. In their role as guides, lncRNAs direct ribonucleoprotein (RNP) complexes to specific genomic sites [28]. For example, Xist’s RepA region recruits the PRC2 complex to the X chromosome to initiate inactivation [35]. Lastly, lncRNAs like HOTAIR and NEAT1 act as scaffolds by binding different regulatory factors and RNA-binding proteins (RBPs) in coordinating gene repression across the genome and driving the formation of stress-induced nuclear bodies [36,37].

Importantly, these four archetypes are not mutually exclusive [28]. A single lncRNA may function in multiple roles simultaneously or in different contexts, illustrating a combinatorial modality that enhances the complexity and specificity of its regulatory function [28]. This multifaceted capacity underscores the diverse and dynamic roles that lncRNAs play in regulating gene expression and cellular function.

## 4. LncRNAs in Human Diseases

The diverse modes of lncRNA action suggest their involvement in many fundamental cellular pathways. Consequently, dysregulated lncRNA expression has been linked to the development of various diseases. This section discusses representative lncRNAs and their roles in disease pathogenesis.

### 4.1. LncRNAs in Cancer

Due to their vital regulatory functions, lncRNAs play critical roles in cancer biology, including regulating tumor growth, invasion, and drug resistance [19,38,39]. LncRNAs function through diverse molecular mechanisms. One such mechanism is chromatin regulation in *cis* [6]. For example, in hepatocellular carcinoma (HCC), the lncRNA lncTCF7 recruits the SWI/SNF chromatin remodeling complex to the TCF7 promoter, activating Wnt signaling and promoting liver cancer [40,41]. In contrast, some lncRNAs act in *trans* to regulate chromatin at distant genomic loci [6]. A well-known example is HOTAIR, which contributes to breast cancer metastasis by directing the PRC2 complex to epigenetically silence genes in the HOXD cluster [6,20,42].

LncRNAs also function at the post-transcriptional level. MALAT1, for instance, regulates alternative splicing by modulating the activity of serine/arginine-rich (SR) splicing factors, influencing gene expression patterns involved in cancer progression [6,43]. Another mechanism involves lncRNAs acting as microRNA sponges [6]. For example, FAM83H-AS1 promotes prostate cancer growth and cell cycle progression by sequestering miR-15a, leading to increased expression of CCNE2 [44]. See Table 1 for additional lncRNAs implicated in tumorigenesis.

Through diverse mechanisms, lncRNAs can act as either oncogenes or tumor suppressors, depending on their specific targets and disease context. Their functional versatility and central involvement in cancer signaling make them compelling subjects for ongoing research.

### 4.2. LncRNAs in Other Diseases

Beyond cancer, lncRNAs act as key regulators in the progression of various diseases. In neurological disorders, SNHG1 modulates neurotoxicity by interacting with microRNAs and has been implicated in the pathogenesis of Parkinson’s disease [50]. Similarly, the expression of NAT-RAD18 exhibits an inverse correlation with that of Rad18, a DNA-repair factor, suggesting a potential role in Alzheimer’s disease [51].

In cardiovascular disease, lncKCND1 binds and upregulates YBX1, promoting cardiac hypertrophy [52], while TUG1 may drive atherosclerosis by sponging miR-145-5p to increase FGF10 expression [53]. In autoimmune disorders, GAS5 regulates inflammatory signaling and cytokine expression and is linked to systemic lupus erythematosus (SLE) [55], whereas promoter SNPs in SAS-ZFAT are associated with autoimmune thyroid disease [56]. In aging, ANRIL recruits the PRC2 complex to repress p15 (INK4B) expression, implicating it in cellular senescence and age-related pathophysiology [58] (Table 1).

### 4.3. Therapeutic Applications of lncRNAs

Given their roles in diverse biological pathways, several lncRNAs have shown promise as biomarkers for cancer progression. For example, the lncRNA PCA3 has been approved by the FDA for the early detection and management of prostate cancer [59,60,61].

Antisense oligonucleotides (ASOs) remain a widely used approach for disrupting lncRNA function [62]. ASO-based strategies targeting oncogenic lncRNAs such as MALAT1 in breast and lung cancer have been shown to suppress tumor progression [63,64]. More recently, CRISPR/Cas-based genome editing tools have emerged as promising alternatives.

Emerging therapies exploit their tumor-specific expression to improve treatment precision and limit off-target effects [6]. One example is BC-819 (DTA-H19), a plasmid-based therapy that uses the H19 promoter to drive diphtheria toxin expression selectively in tumor cells. This approach has completed a Phase 1/2a clinical trial, showing reduced tumor size across carcinomas [65]. Similarly, Xist has been engineered to silence an extra copy of chromosome 21 in trisomic cells derived from individuals with Down syndrome [66]. These studies underscore the versatility of lncRNA-based therapeutic strategies and support the continued investigation of their clinical applications.

## 5. Technologies for lncRNA Functional Characterization and Validation

While lncRNAs are garnering significant interest as potential therapeutic targets for various diseases, including cancer, the mechanisms by which they exert their functions remain largely unexplored. To date, only a limited number of lncRNAs have been extensively studied, despite an estimated 20,000 lncRNAs being annotated in the human genome [9]. As one of the promising strategies for studying lncRNAs at the molecular level, this paper focuses on their structure-function relationships.

### 5.1. Structure-Function Relationships of lncRNA

By revealing the structural and mechanistic basis of biomolecular function, structural biology has greatly advanced therapeutic discovery and design. Three-dimensional protein structures illuminate ligand binding and self-assembly, deepening our understanding of molecular mechanisms. Extending these insights to nucleic acids, studies of RNA and DNA structures have proven to be central to understanding biological processes. For example, the double-helical structure of DNA revealed how genetic information is stored, replicated, and transmitted with high fidelity [67].

The development of advanced RNA structural determination techniques has enabled the determination of RNA structures, including those of riboswitch RNAs and self-splicing introns [68,69,70]. With growing recognition of RNA’s diverse roles in gene regulation and disease, structural characterization of RNA molecules will remain central to both fundamental discovery and the development of RNA-centric therapies.

In general, RNAs form versatile structures through canonical and non-canonical base-pairings as well as non-specific backbone-to-base and backbone-to-backbone interactions [71]. Because of these physical properties and their length, most lncRNAs can fold into complex secondary and tertiary structures. These secondary structural elements are often evolutionarily conserved and functionally relevant [35,72]. For example, the lncRNAs roX1 and roX2 assemble the male-specific lethal (MSL) dosage compensation complex in *Drosophila melanogaster*, via multiple tandem stem-loop motifs as a binding platform [73]. The lncRNA MEG3 is another example that well-illustrates the functional significance of lncRNAs’ structural motifs. The tumor suppressor MEG3 has conserved, structured domains. Mutations that disrupt the long-range, pseudoknot interaction between its two distal structural motifs destroy MEG3-dependent p53 stimulation, whereas mutations that retain the secondary structure remain functional [74].

Although these examples have often been used to demonstrate that lncRNAs’ functions are more dependent on structural conservation than on sequence conservation, Rivas et al. (2017) reported that their R-scape method did not detect conserved secondary structure in several lncRNAs, indicating possible sequence-based functions in some lncRNAs [75]. These findings highlight the difficulty of drawing definitive conclusions about the relative contributions of structure versus sequence, in part because high-resolution structural data are available for only a limited number of RNAs. As additional genomes are sequenced and comparative datasets expand, conserved features—whether structural or sequence-based—may become more readily detectable across lncRNAs. Nonetheless, accumulating evidence supports the conservation of structural elements in lncRNAs [76,77].

### 5.2. Hierarchical Nature of lncRNA Structural Complexity

#### 5.2.1. The Primary Structure of lncRNA and Its Functional Significance

LncRNAs function at structural complexity levels ranging from primary structure to tertiary structure. On one end, there are lncRNAs whose mechanisms are based on their primary structures. The primary structure, defined as the linear nucleotide sequence of lncRNA, can be determined using standard RNA sequencing and can itself have functional significance. For example, lncRNA 1/2-sbsRNA forms Alu element–mediated duplexes with complementary sequences in the 3′ UTR of target mRNAs, creating a double-stranded structure recognized by STAU1, which triggers Staufen-mediated mRNA decay (SMD) [78]. Another way through which lncRNAs exert translational control is by sponging miRNAs that would otherwise sequester and regulate the translation of target mRNAs (Figure 2A) [79,80].

#### 5.2.2. The Secondary Structure of lncRNA and Its Functional Significance

Some lncRNAs function by adopting specific secondary structures. Secondary structure refers to local base-pairing interactions within RNA that fold a linear sequence into motifs such as stem-loops, multi-way junctions, and bulges. These structures can be dynamic and provide platforms for interactions with proteins and other nucleic acids, while also contributing to higher-order three-dimensional RNA architecture. Their functional relevance is typically assessed by combining structural characterization with cellular or biochemical assays. For example, the independent secondary-structure modules of HOTAIR are crucial for binding to multiple chromatin-modifying proteins, including PRC2 [81]. Additionally, the A-repeat region of Xist forms multiple helices, providing a structural scaffold for the binding of functional proteins such as SPEN [82]. More recently, the secondary structure of the lncRNA MUNC was determined (Figure 2B), revealing multiple structurally defined domains that function independently to regulate distinct gene programs during myogenesis, likely through interactions with different protein partners [83]. Additionally, lncRNA SChLAP1 was shown to harbor functionally important terminal structural modules that promote cancer cell proliferation and invasion [84,85].

#### 5.2.3. The Tertiary Structure of lncRNA and Its Functional Significance

In contrast, some lncRNAs depend on complex tertiary structures for function. These higher-order three-dimensional architectures involve long-range interactions between distant RNA regions. The crystal structure of the 3′ end of MALAT1, for example, revealed a triple-helix motif that stabilizes the transcript by protecting it from rapid decay (Figure 2C) [86]. Recently, the low-resolution tertiary structure of full-length Braveheart was revealed utilizing SAXS, showing that it is made up of three structured domains that cooperatively interact with CNBP [87]. In addition, HOTAIR and Xist have emerged as prominent models for investigating lncRNA tertiary structure [88,89].

As discussed above, structural features contribute to lncRNA function. However, only a small fraction of the ~20,000 annotated human lncRNAs have been examined using structural approaches. With recent advances in RNA structure determination methods, an increasing number of lncRNAs are now being structurally characterized (Figure 3). The following section discusses these methodological approaches.

### 5.3. Experimental Methods to Determine Secondary Structures of lncRNA

To determine lncRNA secondary structure, several experimental approaches are commonly used. Chemical probing methods modify flexible nucleotides, while enzymatic methods cleave structured or unstructured regions. These modifications are typically detected by reverse transcription followed by high-throughput sequencing. When applying these methods, key factors include probe selection, reverse transcriptase selection, and sequencing strategy to accurately capture nucleotide reactivities.

#### 5.3.1. Probes

Secondary structure studies of lncRNA are an active area of research, primarily focusing on inferring single- and double-stranded regions of RNA. Since each probe reacts differently with RNA, mixing and matching various probes can yield a more accurate secondary structural map. There are two main types of probes: enzymatic and chemical.

***Enzymatic probes.*** Enzymatic probes include RNases with different reactivities toward RNA. RNase V1 digests double-stranded RNA regions, whereas RNase S1 and RNase P1 specifically cleave single-stranded regions. Combined with high-throughput sequencing for the digested fragments, these enzymatic footprinting techniques can map single- and double-stranded regions of RNA [90]. The secondary structures of lncRNAs roX1 and roX2 were determined utilizing enzymatic probing [73]. Unfortunately, enzymatic probing is highly sensitive to steric hindrance. Additionally, since RNases cannot penetrate cellular membranes, they are applicable to in vitro structural studies, and not in cellulo [91].

***Chemical probes.*** Chemical probes involve RNA-modifying chemicals (Table 2). There are probes with base-specificity (Figure 4A). For instance, dimethyl sulfate (DMS) methylates mainly unpaired A and C, while 1-cyclohexyl-3-(2-morpholinoethyl) carbodiimide metho-*p*-toluene sulfonate (CMCT) acylates unpaired U and G [92,93]. There are other probes, such as 2-keto-3-ethoxy-butyraldehyde (kethoxal) and diethylpyrocarbonate (DEPC), which modify single-stranded G and A, respectively [94,95].

Secondary structure probing with traditional probes can be less efficient due to bias in base specificities and the need for multiple enzymatic reactions. Recently, chemical probes without base-specificity have become widely used, streamlining the structure determination processes. They react with the sugar-phosphate backbone (Figure 4A) [100]. Typical examples are SHAPE (selective 2′-hydroxyl acylation and primer extension) reagents such as N-methylisatoic anhydride (NMIA) and 1-methyl-7-nitroisatoic anhydride (1M7), each with distinct half-lives that allow for RNA structural studies over different time scales [99,100,116]. This is beneficial when RNAs have multiple transition states with functional significance. This is particularly important in cellular environments where RNAs gradually form structures co-transcriptionally and interact with binding partners. Most SHAPE probes, including 2-methylnicotinic acid imidazolide (NAI), are suitable for RNA structural probing in vitro and in cellulo [103]. An increasing number of SHAPE reagents are being developed, such as 5-nitroisatoic anhydride (5NIA), 2-methyl-3-furoic acid imidazolide (FAI), and 2-aminopyridine-3-carboxylic acid imidazolide (2A3) [103,105,107]. Newer SHAPE reagents provide improved signal-to-noise ratios, cellular permeability and performance in cellular environments, expanding the toolkit for RNA structure determination.

***Chemical probing in cellular environments.*** In cellulo chemical probing is increasingly important for functional structure determination, particularly for identifying stable motifs and interaction sites on lncRNA molecules. RNA folding under physiological conditions is influenced by RNA chaperones as well as interacting partners, such as proteins, DNA, and other RNA species [4,117,118]. As such, a lack of reactivity in cellulo does not necessarily indicate a double-stranded region. Non-reactive sites may also result from protection by proteins or other nucleic acids, or from intra- or intermolecular interactions that block probe access [119,120,121].

To identify direct binding sites on lncRNAs in cellulo, a novel technique called RNP-MaP (ribonucleoprotein networks analyzed by mutational profiling) has recently been developed [122]. In this process, cells are treated with NHS-diazirine (SDA), whose succinimidyl esters react with the lysine side chains of interacting proteins. The SDA-treated and non-treated cells are then exposed to UV light. UV-activated diazirine reacts with RNA nucleotides, labeling protein binding sites on lncRNAs. Subsequently, the cells are lysed, and crosslinked proteins are digested by proteinase K treatment. The remaining small peptides after digestion act as bulky adducts, introducing mutations during reverse transcription. After sequencing, the reads are mapped to the full-length lncRNA sequence, allowing for the localization of direct protein-binding sites on the lncRNA within cells [122].

#### 5.3.2. Reverse Transcriptase

Selecting an optimal reverse transcriptase is crucial for structure probing, as the current readout heavily relies on high-throughput sequencing, where cDNA synthesis is key. An efficient reverse transcriptase should synthesize cDNA smoothly, without natural pausing, and accurately drop off or incorporate mutations at modification sites. For this purpose, the SuperScript reverse transcriptase family (Invitrogen) has been widely used. These enzymes demonstrate lower fidelity of reverse transcription (RT) reaction in the presence of Mn^2+^, introducing random nucleotides at the chemically modified sites [123].

***Reverse transcription (RT) of long RNA.*** Recently, highly processive reverse transcriptases have been developed. These include thermostable group II intron reverse transcriptase, TGIRT, and group II intron maturase of *Eubacterium rectale*, MarathonRT (MRT) [124,125,126]. In contrast to conventional reverse transcriptases such as SuperScript II, which often fail to produce long amplicons, these novel enzymes can generate long cDNA products in a single pass, enabling detection of all modifications across the RNA strand (>1 kb) in a single analysis [126]. These novel reverse transcriptases enable accurate secondary structure prediction of long RNAs.

***Post-transcriptional RNA modification detection.*** Reverse transcriptase has been further adapted for multiple purposes beyond synthesizing cDNA from long RNAs. For instance, MRT, when combined with mutational profiling, has been utilized to detect chemical modifications in RNA [127,128].

Thus far, a wide range of post-transcriptional modifications has been characterized, including pseudouridine (Ψ), N6-methyladenosine (m6A), and 5-methylcytosine (m5C). These RNA modifications play crucial roles in cellular functions by regulating gene expression, and they can significantly alter RNA structures, ultimately affecting RNA-protein interactions [129]. For example, HNRNPC binds and stabilizes mRNA to promote mRNA processing [130,131,132,133]. The U-tract sequences on its target mRNA, where HNRNPC binds, are buried in a stem-loop structure, thereby inhibiting the binding of HNRNPC to this region [134]. However, when adenosines in the stem-loop are methylated (m6A), the tight stem-loop structure destabilizes, exposing the U-tract region [134]. Therefore, HNRNPC could readily bind to the U-tract region [134]. Taken together, linking RNA modifications and structures will allow for a deeper understanding of RNA’s functional mechanisms and significance.

Recently, MRT-ModSeq, which combines next-generation sequencing and machine learning, has been developed to detect multiple types of RNA modifications [128]. Overall, the choice of reverse transcriptase is critical not only for resolving RNA secondary structure but also for enabling the detection of RNA modifications. Here, we highlighted this technical aspect in the context of structural probing, whereas the biological implications of RNA modifications are examined separately in Section 8.2.

#### 5.3.3. Reactivity Readout

Traditionally, gel- and capillary-electrophoresis–based methods were used to detect enzymatic or chemical modifications in RNA, yielding nucleotide reactivity profiles. More recently, these approaches have been replaced mainly by high-throughput next-generation sequencing (NGS). By enabling parallel analysis of multiple RNA species in a single experiment, NGS-based methods support transcriptome-scale determination of RNA secondary structure. At present, the combination of chemical probing with high-throughput sequencing is widely used to infer RNA secondary structure. The general principle underlying these approaches is the detection of reverse transcriptase (RT) stops or mutation events, which are mapped back to a reference RNA sequence to generate reactivity profiles.

***Reverse-transcriptase termination:*** In these methods, reverse transcriptase prematurely terminates cDNA synthesis upon encountering chemically modified nucleotides, generating truncated cDNA fragments that are subsequently deep sequenced (reviewed in [135]). Interpretation requires distinguishing true modification-induced termination events from natural RNA ends, a challenge that becomes more pronounced for long RNAs due to RNA degradation and incomplete reverse transcription [135].

***Reverse-transcriptase mutation:*** Mutational profiling (MaP-Seq) was developed to address several limitations of RT termination strategies (Figure 4B). Under defined reaction conditions—such as substituting Mn^2+^ for Mg^2+^—reverse transcriptase fidelity is reduced, causing chemically modified nucleotides to be misread and encoded as mutations in the resulting cDNA [123]. Mapping these mutations to the reference RNA generates reactivity profiles, in which elevated mutation frequencies correspond to flexible or unpaired regions, while reduced mutation frequencies indicate base-paired or structurally constrained regions [102,136].

While MaP-Seq has gained broad adoption, RT termination–based methods remain valuable. Each approach exhibits distinct biases, and together they can provide complementary insights into RNA secondary structure [137].

#### 5.3.4. Limitations and Breakthroughs

Since the tools for determining RNA secondary structure, as discussed above, primarily focus on identifying whether nucleotides are single-stranded or double-stranded, subsequent structure modeling based on nucleotide reactivities is required [102,136]. The reactivities represent an averaged profile of potential alternative structures, if present. Moreover, the structures of lncRNA are complicated due to repetitive sequences, long-range intramolecular interactions, and intermolecular interactions with binding partners. This complexity often results in the generation of multiple structural models, making structural analysis more challenging [138]. Taken together, structural studies of lncRNAs must address the difficulties arising from their large size and structural heterogeneity.

***Large-sized lncRNA.*** To address the large size of lncRNA, the 3S (Shotgun Secondary Structure) approach can be utilized, where a full-length lncRNA is fragmented into shorter segments [139]. The fragments will be probed, and then, their reactivities will be matched with the reactivity of a full-length lncRNA [139]. High correlation indicates the presence of independently folding domains that fold locally, whereas low correlation indicates long-distance interactions [139].

While 3S applies conventional chemical probing to fragmented RNAs, long-range probing methods are actively being developed to directly detect base-pairings and multiple conformations. By incorporating crosslinking steps into the protocol, PARIS (psoralen analysis of RNA interactions and structures) can identify direct base-pairing interactions and alternative structures [82]. Other methods using psoralen and its derivatives include SPLASH (sequencing of psoralen cross-linked, ligated, and selected hybrids) and COMRADES (cross-linking of matched RNAs and deep sequencing [140,141]). They have unique strategies to increase the efficiency of psoralen crosslinking. Finally, another method, RING-MaP (RNA interaction groups by mutational profiling), analyzes correlated positions of chemical modifications to identify base pairs [142].

Notably, researchers are revisiting terbium (III) (Tb^3+^) as a chemical probe to detect local tertiary structures of large RNAs. When RNA folds into compact tertiary structures, negatively charged phosphate groups are positioned closely together, creating localized regions of high negative charge [112,113]. These sites recruit multivalent cations such as Mg^2+^, which reduce electrostatic repulsion and stabilize the fold [112,113]. Lanthanide ions, including Tb^3+^, also bind to these negatively charged regions, but in contrast to Mg^2+^, Tb^3+^ promotes RNA cleavage [114]. The resulting cleavage pattern marks local tertiary structures near the bound sites [114]. Combined with deep sequencing, Tb-seq maps these tertiary motifs and enables targeted functional or high-resolution structural analyses [115].

***Structural heterogeneity.*** Cellular RNA exhibits structural heterogeneity due to alternative processing and inherent structural dynamics. This is well-illustrated by eight distinct structural models proposed for the Xist A-repeat, each derived using different techniques [35,82,89,143,144,145,146,147,148,149,150]. lncRNA can exist as dynamic conformations in a physiological condition, changing the abundance of specific conformations according to environmental stimuli [151]. Such conformational changes can regulate cellular function. For example, riboswitches alter their structure in response to small molecules, thereby controlling gene expression. For a comprehensive review on RNA structural dynamics, see Spitale and Incarnato (2023) [152].

To address the structural heterogeneity of lncRNAs, several algorithms for single-molecule RNA structural analysis have been developed (Figure 5A). They focus on a single RNA molecule to model its structures, whereas current structure probing methods create a structural model based on the population average of RNA. For example, the DaVinci algorithm showed that lncRNA COOLAIR exists in diverse conformations in vivo, with these variations influencing gene expression in response to environmental stimuli [153]. Similarly, the DREEM algorithm demonstrated the structural heterogeneity of HIV RNA in vivo and its functional significance [154].

To address the challenges posed by the large size and heterogeneity of lncRNAs, long-read sequencing technologies have recently gained significant attention. While second-generation platforms such as Illumina are widely used, their short reads limit the analysis of full-length and repetitive lncRNAs [155,156]. Third-generation approaches, including nanopore sequencing (Oxford Nanopore Technologies), overcome these limitations by reading individual RNA molecules in full length. As each nucleotide passes through a nanoscale pore, characteristic changes in ionic current allow base identification [157,158,159,160,161]. This enables the sequencing of RNA molecules up to megabase lengths [162,163].

Using this new technology, Nanopore DMS-MaP (Nano-DMS-MaP) was developed to determine isoform-specific RNA structures [164] (Figure 5B). Following DMS probing, long cDNAs are synthesized using an ultra-processive reverse transcriptase, and nanopore sequencing is then applied to analyze full-length transcripts—something not achievable with short-read methods. Using Nano-DMS-MaP, researchers examined human immunodeficiency virus (HIV)-1 transcripts and discovered that spliced isoforms contain unfolded regions within the 5′ untranslated region (UTR) that are absent in unspliced transcripts. These structural differences occur near the packaging site, suggesting a mechanism by which spliced viral RNAs are excluded from virions [164].

Moreover, Nanopore sequencing can detect a broader range of RNA modifications than current methods, which are largely limited to only a few of the most common modifications, such as m6A and 5mC [165]. Additionally, Nanopore sequencing enables direct sequencing of native RNA strands without the need for RT or PCR amplification, making it a promising tool for studying lncRNA, which often has low expression levels [165].

Overall, the inherent flexibility of lncRNA poses a significant challenge to their structural determination, necessitating an integrative approach that combines various experimental methods that address the dynamic, polydisperse conformations lncRNA may adopt.

### 5.4. Towards Tertiary Structures of lncRNA

While tertiary lncRNA structures remain underexplored, recent advances in secondary structure studies and the development of a wide range of techniques have accelerated efforts in this area [166]. Low-resolution tertiary structure studies benefit from small-angle X-ray scattering (SAXS) and atomic force microscopy (AFM), whilst cryo-electron microscopy (Cryo-EM) and nuclear magnetic resonance (NMR) provide useful tools for high-resolution tertiary structures at an atomic level.

***Cryo-EM.*** In cryo-EM, biomolecules are rapidly frozen and subjected to transmission electron microscopy. Hundreds of thousands of two-dimensional images taken from different orientations are used to reconstruct the biomolecule’s tertiary structure. The high-resolution structure of the ribosome was determined utilizing cryo-EM, and more recently, this technique was applied to study the tertiary structure of PRC2 and the G-quadruplex RNA complex [167].

The PRC2 complex is a chromatin modifier that methylates histone H3 lysine 27 through its enzymatic subunit EZH2 [168]. The 3.3 Å resolution cryo-EM structure of the PRC2-RNA complex revealed the mechanism of RNA-mediated PRC2 inhibition [167]. The G-quadruplex RNA binds and dimerizes PRC2, forming an interface between two EZH2 subunits and blocking nucleosome binding [167]. This illustrates how small RNA domains can mediate interactions with protein partners.

Other chromatin modifiers and transcription factors also interact with RNA, often without canonical RNA-binding motifs [167,169,170]. This highlights the need for structural studies to investigate the detailed regulatory roles of RNA. In this context, cryo-EM has emerged as a leading tool for investigating RNA and RNP complexes due to the recent advances in electron microscopy and computational modeling. Cryo-EM has been applied to large RNP complexes, such as ribosomes, as well as protein-free RNAs, including small riboswitches and portions of SARS-CoV-2 RNA. It also enables analysis of RNAs with dynamic conformations.

***X-ray crystallography.*** In X-ray crystallography, crystallized RNA scatters X-ray radiation, creating a unique scattering pattern, or ‘electron density’, which is used to determine the RNA’s tertiary structure. This method determined the tertiary structure of the 3′-end of MALAT1, revealing how its stable triple-helix structure protects it from rapid degradation [86]. However, this technique is not ideal for determining unstructured and flexible biomolecules, as large RNAs often have flexible regions and multiple conformations [171]. Additionally, RNA is intrinsically more difficult to crystallize than proteins [172]. As a result, engineering the RNA of interest is beneficial for obtaining well-diffracted RNA crystals, as shown by deleting flexible regions that are not functionally important [70]. Crystallization can also benefit from screening and choosing the RNA target, which may fold at lower Mg^2+^ concentrations and adopt a stable conformation more readily, facilitating crystal growth. For example, the tertiary structure of the self-splicing group II intron from *Oceanobacillus iheyensis* was successfully determined by X-ray crystallography [70]. The crystal structure of this ribozyme revealed that its domains and subdomains exhibit extensive tertiary interactions, forming a core with a catalytic triple helix [70]. Ultimately, this cooperatively constructed triplex generates a negatively charged binding pocket for two catalytically important magnesium ions [70].

Recently, cryo-EM structures of group II intron captured at different splicing stages provided a detailed understanding of its catalysis mechanism [173]. This suggests that combining X-ray crystallography with alternative techniques, such as SAXS and cryo-EM, is gaining popularity because they do not require RNAs to be crystallized.

***SAXS.*** Unlike X-ray crystallography, SAXS measures scattering from non-crystallized, randomly oriented molecules in solution. This technique constructs tertiary structures by averaging the sizes and shapes of the molecule, providing an overall structural architecture of lncRNA. For instance, the low-resolution, tertiary structure of the lncRNA Braveheart was determined by SAXS [87]. This, in turn, can serve as a foundation for higher-resolution structure determination using techniques like cryo-EM.

Furthermore, SAXS is ideal for studying lncRNA structures with multiple conformations in solution. Plumridge et al. (2018) utilized SAXS to study RNA structural ensembles during magnesium ion-induced RNA folding of a three-helix junction, where novel folding intermediates were identified [174]. In addition, they coupled SAXS with MD simulation, which benefits structural studies of lncRNA with dynamic conformations. More recently, significant advancements have been made in integrating SAXS experimental data with MD simulations [175].

***Atomic force microscopy (AFM).*** In AFM, a minute tip is brought close to a sample’s surface. The tip, connected to a cantilever, detects Van der Waals forces between the tip and the atoms on the surface of the sample, thereby generating structural data. AFM allows structural analysis of native RNAs in physiological conditions and is a well-suited tool to study flexible structures of lncRNA. For example, in vitro transcribed HOTAIR RNA was examined by AFM, revealing nine structural segments that form well-defined local structures within a flexible global architecture [88]. Dynamic movements of HOTAIR were observed during interactions with genomic DNA, and its flexible nature was further supported by cryo-EM 2D images [88]. AFM can thus complement techniques such as cryo-EM, although it is less commonly used due to its lower resolution.

***Nuclear Magnetic Resonance Spectroscopy (NMR).*** In NMR, nuclei absorb electromagnetic radiation and transit to different energy levels in a strong magnetic field. Each nucleus has a unique resonance frequency in different chemical environments, allowing NMR to obtain the molecule’s structural information. Recently, NMR was used to investigate individual exons of HOTAIR, elucidating its intrinsic flexibility [176]. This finding aligns well with previous secondary structure probing data, demonstrating NMR’s utility in lncRNA with multiple conformations and transition rates. Although NMR can apply only to small RNA motifs, and it has limitations on buffer conditions, it can become a useful tool when combined with other biophysical techniques. For instance, structural studies with NMR and X-ray crystallography revealed how m6A modifications of the XIST A-repeat are recognized by the YTHDC1 protein and ultimately lead to conformational changes that alter the accessibility of additional RNA-binding proteins [177]. The A-repeat has conserved AUCG tetraloops with m6A modifications [178]. X-ray crystallography indicated that YTHDC1 binds the (m6A)UCG-tetraloop in a manner consistent with single-stranded recognition, while the lower stem-loop remains base-paired [177]. NMR confirmed that YTHDC1 binding locally destabilizes the upper stem, partially loosening the hairpin without full disruption. This partial destabilization further facilitates YTHDC1 binding and modulates the accessibility of additional RNA-binding proteins [177].

## 6. LncRNA Structures as Therapeutic Targets

***Small-Molecule Targeting of RNA Structures*.** LncRNAs with well-defined biological functions can serve as therapeutic targets. Therapeutic strategies include directly targeting structured RNA regions or disrupting lncRNA-protein interactions.

MALAT1 provides a well-characterized example of an lncRNA that can be directly targeted by small molecules. Small-molecule ligands have been shown to bind a structured region at the 3′ end of MALAT1 that regulates RNA stability [179]. MALAT1 is particularly well suited for such approaches due to its stable and unique 3′ structural motif. Accordingly, techniques such as SHAPE-MaP are essential for defining structured and targetable RNA motifs.

A major limitation of structure-based RNA targeting is the scarcity of high-resolution three-dimensional structures for lncRNAs. RNA structural dynamics present an additional challenge, as many RNAs adopt multiple conformations rather than a single stable state [171]. Identifying the functional conformation and its associated protein partners, therefore, remains difficult. For example, the A-repeat region of Xist exhibits substantial structural heterogeneity. In such cases, small-molecule strategies often focus on disrupting lncRNA interaction networks rather than targeting a single defined structure. Consistent with this approach, small molecules that bind Xist reduce the conformational freedom and suppress breast cancer growth and metastasis by interfering with PRC2 interactions [180]. To better address RNA conformational heterogeneity, hybrid strategies combining experimental methods with molecular dynamics (MD) simulations are being developed to characterize RNA structural ensembles.

In parallel with structural studies, RNA-focused small-molecule libraries are being generated. These libraries incorporate physicochemical features associated with RNA binding and enable high-throughput screening of lead compounds [181]. Such approaches increasingly integrate rational design strategies, further accelerated by advances in artificial intelligence (AI). Expanding RNA structural datasets now allow machine-learning (ML) models to analyze large chemical libraries, identify RNA-binding chemotypes, and optimize lead compounds. For example, the Hargrove laboratory employs ML-based QSAR models trained on in vitro datasets to evaluate physicochemical and structural features associated with RNA affinity [182].

## 7. RNA Structure and Its Protein Interactome

LncRNA function depends heavily on RNA-protein interactions [118,183,184]. These interactions occur through several modes: recruitment of proteins to genomic regions (guide), assembly of ribonucleoprotein (RNP) complexes (scaffold), or sequestration of proteins (decoy) [183].

Adaptor proteins represent a subclass of RNA-binding proteins RBPs that link lncRNAs to large protein complexes and broaden the range of possible interactions [185]. Both RBPs and lncRNAs show functional diversity and often exhibit broad interaction profiles rather than strict specificity [186]. Classic examples include the PRC2 chromatin-modifying complex, whose binding to specific lncRNAs is debated [187,188]. More recent evidence has emerged for adaptor proteins that mediate specific interactions between lncRNAs and chromatin remodeling complexes [41]. Mapping the full set of lncRNA-protein interactions and identifying functionally essential pairs is therefore a key objective.

### 7.1. lncRNA-Protein Interaction in Gene Regulation

Many lncRNAs are associated with chromatin-modifying complexes such as SWI/SNF, which regulate both gene activation and repression by altering nucleosome positioning [18,189,190]. lncTCF7 recruits the SWI/SNF complex to activate TCF7 transcription, thereby stimulating WNT signaling and cancer progression [40,41]. In contrast, UCA1 binds SWI/SNF and inhibits its chromatin association at the p21 promoter, promoting cancer cell proliferation [191].

These examples, among many others, highlight the need to determine how lncRNAs modulate chromatin remodeling. Identifying direct binding partners and defining the corresponding binding sites are essential for understanding lncRNA-mediated gene regulation.

### 7.2. Methods to Analyze lncRNA-Protein Interactions

Methods to study lncRNA-protein interactions include protein-centric and RNA-centric approaches. Protein-centric methods, such as RNA immunoprecipitation (RIP) and cross-linking and immunoprecipitation (CLIP), identify lncRNAs bound to specific proteins [192,193]. RNA-centric methods use the lncRNA as bait to isolate interacting proteins. Here, we focus on RNA-centric approaches to systematically uncover proteins that associate with a specific lncRNA. While protein-centric methods are widely used to discover RNAs bound to a given protein, RNA-centric strategies are necessary to determine which proteins interact with a particular lncRNA, providing direct insight into its functional roles. For a comprehensive overview of RNA-protein interaction methods, including protein-centric approaches, see Ramanathan et al. (2019) [192].

Among in vitro RNA-centric approaches, biotinylated RNA pulldown is widely used. Biotin-tagged in vitro transcribed RNA is incubated with cell lysate, and associated proteins are captured by streptavidin beads and identified by mass spectrometry. This method is rapid and compatible with mutagenesis studies but does not reflect native RNA modifications or cellular structural context [192].

In vivo methods such as RNA antisense purification coupled with mass spectrometry (RAP-MS) and comprehensive identification of RBPs by mass spectrometry (ChIRP-MS) address these limitations [194,195]. RAP-MS uses long biotin-labeled DNA probes (~120 nt) and UV crosslinking, increasing specificity and detecting primarily direct interactions [192,194]. ChIRP-MS uses short probes (~20 nt) and formaldehyde crosslinking, capturing both direct and indirect interactions [192,195]. Both methods require large cell numbers due to low recovery after crosslinking and hybridization (Table 3) [192].

Newer techniques aim to overcome these limitations. Protein-crosslinked RNA extraction (XRNAX) isolates UV-crosslinked RNA-protein complexes from the interphase of acidic guanidinium thiocyanate–phenol extraction (TRIzol) (Table 3 and Figure 6A) [196,197]. XRNAX allows global discovery of RNA-binding proteins, and, when combined with immunoprecipitation, can detect specific interactions [197].

**Table 3 cells-15-00105-t003:** The required cell quantity for selected RNA-protein interactome studies.

Method	Application	Cell Number	Reference
RAP-MS	Xist, SARS-CoV-2 viral RNA	200–800 million	[198,199]
ChIRP-MS	Xist	100–500 million	[200]
XRNAX	RNA-bound proteome	10–100 million	[197,201,202,203,204]
O-MAP	47S pre-ribosomal RNA, lncRNA Xist	0.35 million	[205]
TREX	NORAD lncRNA, U1 snRNA, 45S rRNA	10–100 million	[206]

Targeted RNase H-mediated extraction of crosslinked RBPs (TREX) improves on XRNAX by using tiling antisense DNA oligonucleotides and RNase H digestion to release proteins bound to defined RNA regions, enabling region-specific interactome mapping (Figure 6B) [206].

Oligonucleotide-mediated proximity-interactome MAPping (O-MAP) is a proximity-labeling method (Figure 6C) [205]. Cells are formaldehyde-fixed, antisense DNA oligos anneal to the RNA, and HRP-conjugated probes bind these oligos to biotinylate nearby proteins and nucleic acids. Biotinylated molecules are directly captured with streptavidin beads and analyzed by mass spectrometry or sequencing [205,207,208].

### 7.3. Applications: RNA-Protein Interactomes in Disease

Interactome studies of lncRNAs have identified protein partners associated with tumor progression and other disease processes [209]. In colorectal cancer (CRC), ChIRP-MS identified hnRNPA1 as a binding partner of SNHG6, which regulates alternative splicing of PKM2 transcripts and glucose metabolism [210]. Using the same technique, CYTOR was shown to form a complex with NCL and Sam68 to activate NF-κB signaling and drive epithelial–mesenchymal transition (EMT) and metastasis [211]. In non–small cell lung cancer (NSCLC), lncRNA TRIDENT binds TRIM28 and enhances its phosphorylation, promoting cell proliferation [212].

Interactome analysis further reveals the functional diversity of lncRNAs in cancer. DUBR acts as a nuclear scaffold for the NuRD complex and mediates gene repression in CRC [213]. LncRNAs also regulate protein localization. LUCAT1 binds HSP90 and retains it in the nucleus, driving constitutive activation of STAT3 in hepatocellular carcinoma (HCC) [214].

A single lncRNA can exert distinct biological effects through different protein partners. LncRNA AC245100.4 interacts with proteins in the p38-MAPK pathway to promote p38 phosphorylation and cell migration in prostate cancer (PCa) [215]. The same lncRNA also binds HSP90, stabilizes IκB kinase, and activates NF-κB signaling in PCa cells [216].

Some lncRNAs have tumor-suppressive roles. LINC02487, for example, inhibits EMT and metastasis in oral squamous cell carcinoma by binding USP17 and blocking SNAI1 deubiquitylation [217]. RUVBL1-AS1 binds VCP and reduces its expression, inducing G_2_/M arrest in HER2^+^ breast cancer [218].

In addition to cancer, lncRNAs have been demonstrated to play a significant role in other diseases through their interactions with various RNPs [8,219]. For example, ChIRP-MS was used to demonstrate that lncRNA Gm41724 interacts with lamina-associated polypeptide 2α (lap2α) upon pressure overload signals, which in turn increases the expression of Rgs4, a signaling protein for GPCRs that activates cardiac fibroblasts and promotes cardiac fibrosis [220]. In diabetic nephropathy, overexpressed lncRNA, evf-2, has been found to interact with hnRNPU, which increases the transactivation of genes associated with cell cycle re-entry and inflammation, driving podocyte injury [221]. In the case of β-thalassemia, lncRNA, NR_120525, has been shown to inversely regulate the expression of HBG1/2, the gene for HbF, through direct binding with ILF2/3, which inhibits S6 kinase [222]. Decreased HbF presents as increased clinical burden in β-thalassemia patients and thus, targeting NR_120525 has therapeutic potential in alleviating symptoms [222].

## 8. Emerging Themes of lncRNAs

As the field of lncRNAs rapidly expands, so have new ways of studying lncRNAs. While most are beyond the scope of this review paper, here we briefly highlight two new areas of interest in lncRNA research.

### 8.1. RNA-RNA Interactions

While lncRNA-protein interactions are crucial for lncRNAs’ function in the cell, lncRNAs’ interactions with various RNAs such as mRNAs, miRNAs, and other lncRNAs are another key mechanism through which lncRNAs fulfill their biological functions [79,80,223].

Most methods that examine lncRNA-RNA interactome start with a crosslinking step that preserves secondary and tertiary structures of lncRNAs, followed by RNA preparation and high-throughput sequencing. For example, Psoralen Analysis of RNA Interactions and Structure (PARIS) involves a psoralen UV crosslinking step, after which RNA duplexes are purified from a 2D gel, ligated via proximity ligation, prepared into a library, and sequenced via high-throughput sequencing [224]. Methods that utilize psoralen cross-linking, such as PARIS, LIGR-seq, and SPLASH, target double-stranded RNA duplexes as psoralen derivatives intercalate between RNA helices and reversibly fix the base-pairing upon UV irradiation [82,225].

In RNA in situ conformation sequencing (RIC-seq), formaldehyde is used to crosslink protein-RNA and protein–protein interactions to fix RBP-mediated RNA-RNA interactions in the cell [225]. Following the crosslinking step, RNP complexes are isolated from cells, labeled with pCp-biotin at the free 3′ RNA ends, and RNAs of the RNP complexes are ligated with proximity ligation, after which the chimeric RNAs are converted into a library and sequenced [225]. Rather than identifying every RNA-RNA interaction in the cell, RIC-seq focuses on RNA-RNA interactions that are stabilized by RNPs and can be modified to determine the RNA-RNA interactome for specific RNPs.

More recently, kethoxal-assisted RNA-RNA interaction sequencing (KARR-seq) was developed to capture intra- and inter-molecular RNA-RNA interactions independent of RBPs and with high sensitivity for low-expressing transcripts. Such high sensitivity is achieved using N_3_ kethoxal, which tags RNAs with an azide group that crosslinks with DBCO on functionalized dendrimers via “click” reactions [226]. The crosslinked RNAs are then isolated, ligated, and sequenced. Additionally, by varying the size of dendrimers, KARR-seq can detect RNA-RNA interactions over a longer spatial distance than PARIS or RIC-seq [226]. These methods that examine lncRNA-RNA interactome have been instrumental in expanding our understanding of lncRNA structures and their biological functions.

### 8.2. Post-Transcriptional Modification of lncRNAs

Akin to proteins, lncRNAs can be modified post-transcriptionally, and these modifications can alter the structure of lncRNAs and thereby, their biological functions. LncRNA modifications refer to reversible or irreversible covalent changes to the nucleosides that are catalyzed by writer enzymes, recognized by reader enzymes, and removed by eraser enzymes. There are more than 100 characterized RNA modifications, and some of these modifications have been implicated in various disease processes, including cancer [227]. Therefore, interrogating how RNA modifications affect lncRNA structure will further enhance our understanding of the relationship between lncRNA structure and function. For example, RNA modifications such as pseudouridylation of U2 snRNA or 2′-O-methylation of rRNA stabilize the structure of the target RNA and optimize its function in splicing or ribosomal biogenesis, respectively [228,229]. For detailed coverage on this topic, please refer to the reviews by Lewis et al. (2017) [129] and Incarnato and Oliviero (2017) [230].

## 9. Conclusions

Growing evidence links lncRNA structure to function, highlighting the value of structural studies for uncovering key molecular mechanisms. Current findings indicate that lncRNAs are globally flexible and organized into modular domains that act as protein-binding sites and scaffolds. This intrinsic flexibility complicates structural determination, necessitating integrative approaches that capture dynamic and heterogeneous conformations. Nevertheless, recent methodological advances are increasingly addressing these challenges.

Because lncRNAs rarely act alone, interactome analyses are equally important. Their molecular partners can influence lncRNA folding, serving as structural switches that stabilize or remodel specific regions. We also reviewed emerging tools that expand the scope and resolution of lncRNA interactome mapping. Combined structure-focused and interactome-based analyses will be essential for elucidating the mechanisms of still understudied lncRNAs.

Although structural and interactome studies have substantially advanced the field, RNA modifications remain an underexplored layer that can profoundly affect RNA structure and activity. Exploring how these modifications influence lncRNA architecture and interactions will provide further insight into their diverse functional mechanisms.

## Figures and Tables

**Figure 1 cells-15-00105-f001:**
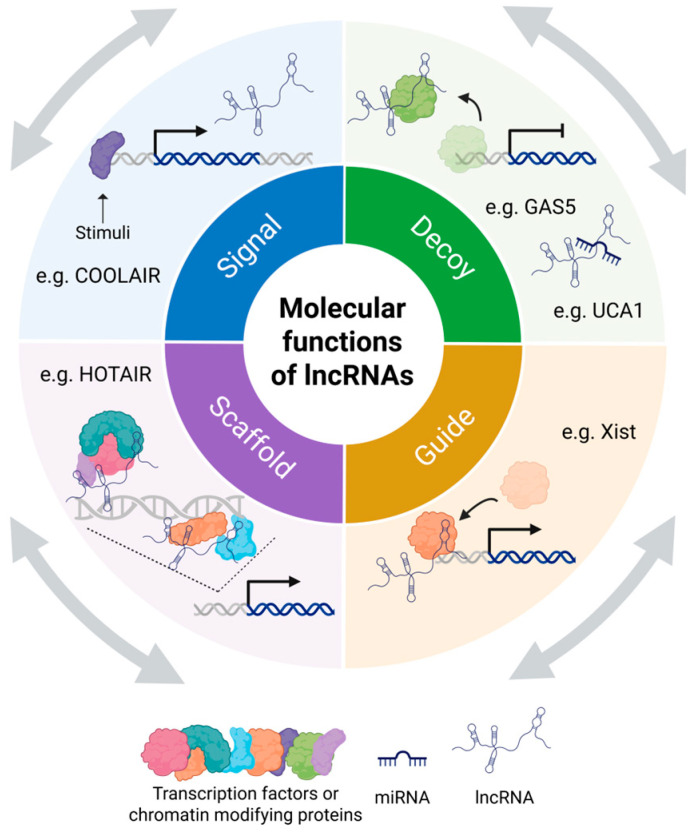
Molecular functions of lncRNAs. Four archetypes of molecular functions that lncRNAs perform include: signals, decoys, guides, and scaffolds [28]. By interacting with various biomolecules, lncRNAs use these archetypes, often in combination, to exert their regulatory roles. Created in BioRender. Oh, M. (2025) https://BioRender.com/y6o1edd (accessed on 12 November 2025).

**Figure 2 cells-15-00105-f002:**
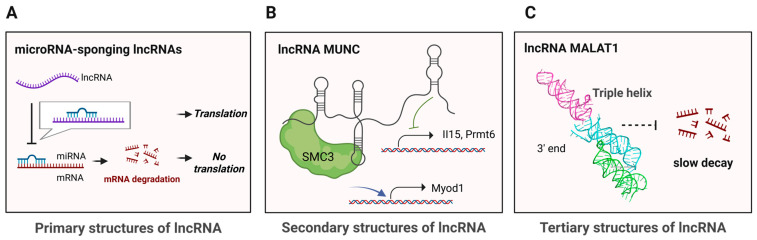
Examples of lncRNAs’ structural complexity levels. (**A**) At the primary structure level, antisense lncRNA complementarily base pairs with corresponding mRNA to dysregulate gene expression. (**B**) At the secondary structure level, lncRNA MUNC has modular domains with distinct functions. (**C**) At the tertiary structure level, lncRNA MALAT1’s triple helix at its 3′ end stabilizes the lncRNA to prevent rapid decay (PBD: 4PLX). Created in BioRender. Oh, M. (2025) https://BioRender.com/j3fkj1g (accessed on 2 December 2025).

**Figure 3 cells-15-00105-f003:**
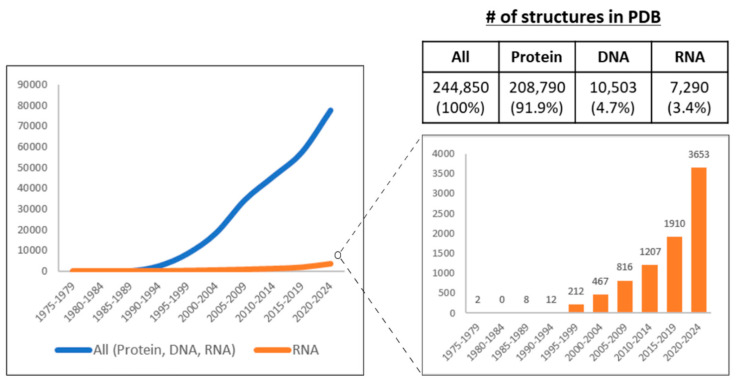
The number (#) of experimentally determined structures of biomolecules in the Protein Data Bank (PDB) from 1975 to 2024. The PDB has been archiving structural data since 1975. The graph presents the data aggregated in four-year intervals. PDB database search was performed for polymer entity types, including protein, DNA, and RNA. All structures were included regardless of refinement resolution. Data from early 2025 are not included in the figure; however, between January 2025 and June 2025, an additional 9561 entries were deposited, comprising 8505, 529, and 527 newly determined protein, DNA, and RNA structures, respectively. Contrary to the exponential expansion of protein structures, structural studies of RNA have lagged (**left**). However, thanks to the recent advances of diverse experimental methods for the structural determination of RNA, the number of RNAs whose structures have been studied has increased, including lncRNAs (**right**). Created in BioRender. Oh, M. (2025) https://BioRender.com/0i4qtl0 (accessed on 11 November 2025).

**Figure 4 cells-15-00105-f004:**
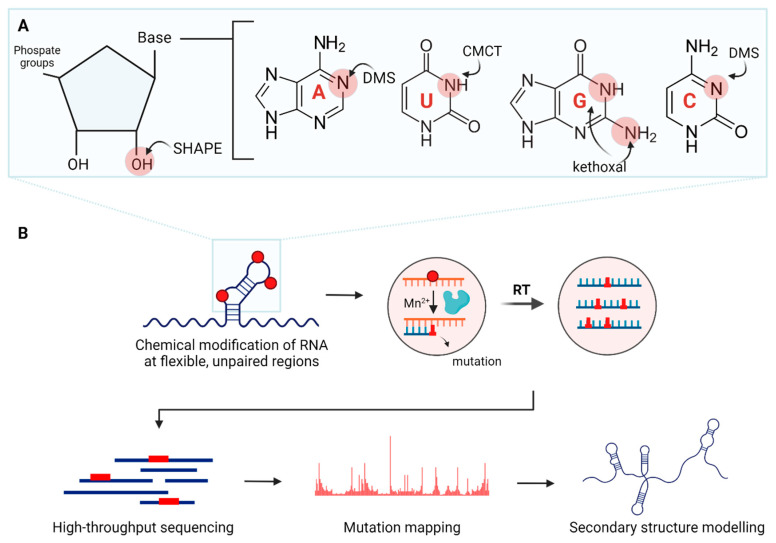
Overview of chemical probing combined with mutational profiling for RNA secondary structural studies. (**A**) Examples of chemical probes with their target areas of modification. (**B**) Chemical probes modify RNA at flexible, unpaired regions. The modifications cause mutations incorporated in newly synthesized cDNA strands during reverse transcription (RT). After high-throughput sequencing, the resulting reads are mapped to the reference RNA sequence to calculate mutation rates. Finally, secondary structural models are determined based on the mutational profiles. Created in BioRender. Oh, M. (2025) https://BioRender.com/lkcu4j9 (accessed on 11 November 2025).

**Figure 5 cells-15-00105-f005:**
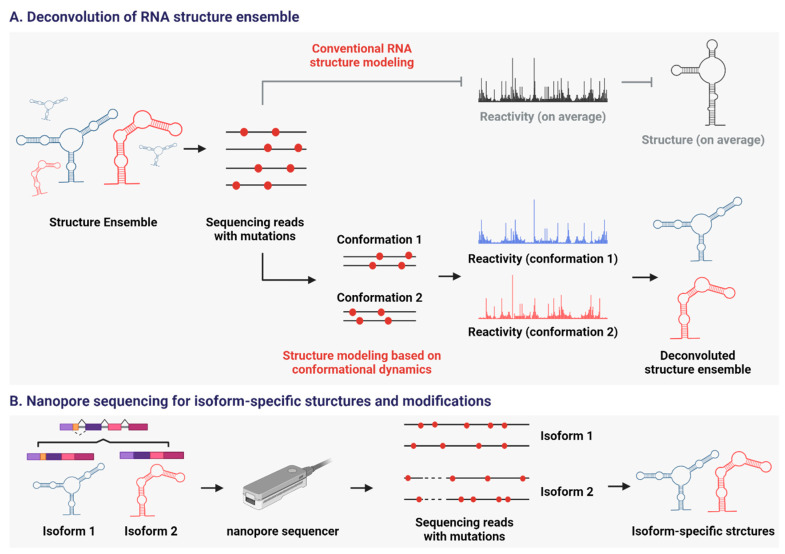
Strategies to deal with the structural heterogeneity of RNA. (**A**) MaP combined with novel algorithms to deconvolute RNA structure ensemble. Their focus is a single RNA molecule, not a population average, to model its structure. (**B**) MaP combined with long-read sequencing. Nano-DMS-MaP can be applied to determine isoform-specific structures and post-transcriptional modifications of RNA. Created in BioRender. Oh, M. (2025) https://BioRender.com/choeb5z (11 November 2025).

**Figure 6 cells-15-00105-f006:**
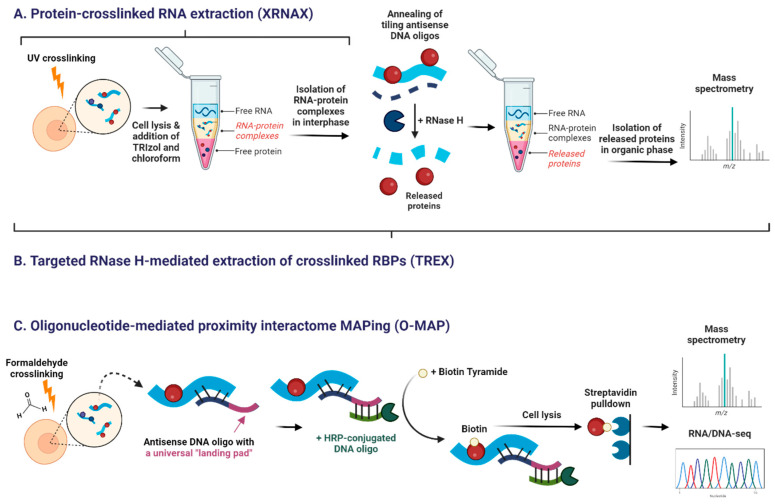
Novel experimental approaches to study RNA interactome. (**A**) Protein-crosslinked RNA extraction (XRNAX) recovers UV-crosslinked RNA-protein complexes in interphase during TRIzol-based RNA extraction. (**B**) Targeted RNase H-mediated extraction of crosslinked RBPs (TREX) isolates crosslinked RNA-protein complexes in a similar way to XRNAX. The RNA of interest is then annealed with tiling antisense DNA oligonucleotides. RNase H specifically digests hybridized RNA, thereby releasing free RNA-binding proteins (RBPs). In the second round of TRIzol-based extraction, the released RBPs are recovered in the organic phase. (**C**) Oligonucleotide-mediated proximity-interactome MAPping (O-MAP) first formaldehyde-crosslinks RAN-protein complexes, followed by annealing of antisense DNA oligos with a universal “landing pad.” The landing pad allows for the annealing of the complementary HRP-conjugated secondary oligos. In the end, the HRP-conjugated oligos biotinylate biomolecules in proximity. After cell lysis, the labeled interactors are recovered by streptavidin pulldown. In all techniques, recovered interactors can be subsequently analyzed by mass spectrometry (MS). Created in BioRender. Oh, M. (2025) https://BioRender.com/s53h2gu (12 November 2025).

**Table 1 cells-15-00105-t001:** lncRNA in human diseases.

lncRNAs in Cancer			
lncRNA	Mechanisms	Diseases	Reference
lncTCF7	Recruits the SWI/SNF chromatin remodeling complex to the TCF7 promoter, activating Wnt signaling and promoting liver cancer	Hepatocellular carcinoma	[40,41]
HOTAIR	Epigenetic modification through the PRC2 complex to silence genes in the HOXD cluster	Multiple cancer	[6,20,42]
MALAT1	Regulates alternative splicing by modulating the activity of serine/arginine-rich (SR) splicing factors, influencing gene expression patterns involved in cancer progression	Multiple cancer	[6,43]
FAM83H-AS1	Sequesters miR-15a, leading to increased expression of CCNE2	Multiple cancer	[44]
UCA1	Regulates the cell cycle through the PI3K/Akt signaling pathway	Bladder cancer	[45]
PCAT14	Modulates chemokines, antimicrobial peptides, and cytokines that influence the infiltration of immune cells.	Prostate Cancer	[46,47]
PCA3	Upregulates AR signaling	Prostate Cancer	[48,49]
**lncRNAs in neurological disorders**			
SNHG1	Regulates neurotoxicity through interaction with microRNAs	Parkinson’s disease	[50]
NAT-RAD18	Shows inverse correlation with Rad18, a DNA-repair regulator	Alzheimer’s disease	[51]
**lncRNAs in cardiovascular diseases**			
LncKCND1	Binds to and upregulates YBX1	Hypertrophy	[52]
TUG1	Sponges miR-145-5p and upregulates FGF10	Hypertension	[53]
SMANTIS	Regulate monocyte recruitment to the vascular wall	Atherosclerosis	[54]
**lncRNAs in autoimmune diseases**			
GAS5	Modulates inflammatory signaling and cytokine expression	Systemic lupus erythematosus	[55]
SAS-ZFAT	Single-nucleotide polymorphisms (SNPs) in its promoter region	Autoimmune thyroid disease	[56]
THRIL	Regulates TNF-α expression through an RNA-protein complex with hnRNPL	Kawasaki disease	[57]
**lncRNAs in aging**			
TERRA	Involved in telomere functions; the mechanism is debated		[7]
ANRIL	Recruits PRC2 complex to suppress the expression of p15(INK4B).		[58]

**Table 2 cells-15-00105-t002:** Chemical probes for determining the secondary structure of lncRNA.

	Target	In Cellulo Probing	Tertiary Structure Probing	Reference
Base specificity				
Dimethyl sulfate (DMS)	A, C, (G)	Yes		[92]
1-cyclohexyl-3-(2-morpholinoethyl) carbodiimide metho-*p*-toluenesulfonate (CMCT)	U, G			[93]
α-ketoaldehydes (e.g., Glyoxal, 2-keto-3-ethoxy-butyraldehyde; kethoxal)	G			[94,96]
diethylpyrocarbonate (DEPC)	A			[95]
carbodiimides (e.g., 1-ethyl-3-(3-dimethylaminopropyl)carbodiimide; EDC, 1-ethyl-3-(3-dimethylaminopropyl) carbodiimide methiodide; ETC)	G, U			[97,98]
**Backbone flexibility (SHAPE)**				
Benzoyl cyanide (BzCN)	2′ OH of ribose			[99]
N-methylisatoic anhydride (NMIA)	2′ OH of ribose			[100]
1-methyl-7-nitroisatoic anhydride (1M7)	2′ OH of ribose	Yes		[101]
1-methyl-6-nitroisatoic anhydride (1M6)	2′ OH of ribose			[102]
2-methylnicotinic acid imidazolide (NAI)	2′ OH of ribose	Yes		[103]
NAI-N3	2′ OH of ribose	Yes		[104]
5-nitroisatoic anhydride (5NIA)	2′ OH of ribose			[105]
2-methyl-3-furoic acid imidazolide (FAI)	2′ OH of ribose	Yes		[103]
FAI-N3	2′ OH of ribose	Yes		[106]
2-aminopyridine-3-carboxylic acid imidazolide (2A3)	2′ OH of ribose			[107]
N-propanone isatoic anhydride (NPIA)	2′ OH of ribose			[108]
**Solvent accessibility**				
Nicotinoyl azide (Naz)	G, A	Yes		[109]
Ethyl nitrosourea (ENU)	Backbone		Yes	[110]
Hydroxyl radical (•HO)	Backbone	(Yes)	Yes	[111]
Metal probes (e.g., Terbium (III) ion, lead (II) ion)	Backbone		Yes	[112,113,114,115]

## Data Availability

No new data were created or analyzed in this study.

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
