# Peer review of "Decoding the lncRNA World: Comprehensive Approaches to lncRNA Structure and Interactome Studies"

_cells, 2026, doi:10.3390/cells15020105_

Round 1
Reviewer 1 Report
Comments and Suggestions for Authors
This review addresses an important and rapidly evolving area of RNA biology and has the potential to serve as a valuable resource for the community. I recommend major revision before publication.
Major concerns:
- The authors claim that antisense lncRNAs regulate translation and provide Figure 2A to illustrate this point. However, translation repression via base-pairing is classically associated with microRNAs, and the reference cited (Ref. 66) does not support translation control by antisense lncRNAs. Instead, Ref. 66 discusses regulation of splicing and mRNA stability, not translational inhibition. This mismatch between the figure, text, and reference needs to be carefully corrected or appropriately supported by an alternative citation.
- RNA–RNA interactions are central to lncRNA biology, yet the manuscript primarily focuses on secondary/tertiary probing and RNP methods. Key technologies developed specifically for mapping RNA duplexes—such as KARR-seq, PARIS, RIC-seq, and related psoralen-based crosslinking approaches—are not discussed. These methods are essential for identifying long-range intramolecular interactions and intermolecular lncRNA–RNA contacts. A dedicated subsection summarizing these tools and their contributions to understanding lncRNA structure–function relationships would substantially strengthen the review.
- The authors should also discuss how post-transcriptional RNA modifications (e.g., m6A, m5C, pseudouridine) reshape lncRNA structure and influence lncRNA–protein interactions. These modifications are increasingly recognized as key regulators of lncRNA function.
Author Response
This review addresses an important and rapidly evolving area of RNA biology and has the potential to serve as a valuable resource for the community. I recommend major revision before publication.
Major concerns:
- The authors claim that antisense lncRNAs regulate translation and provide Figure 2A to illustrate this point. However, translation repression via base-pairing is classically associated with microRNAs, and the reference cited (Ref. 66) does not support translation control by antisense lncRNAs. Instead, Ref. 66 discusses regulation of splicing and mRNA stability, not translational inhibition. This mismatch between the figure, text, and reference needs to be carefully corrected or appropriately supported by an alternative citation.
Answer: We appreciate the reviewer’s insightful comment. We have revised the figure, corresponding text, and supporting references to accurately reflect the regulatory roles of lncRNA primary structure.
- RNA–RNA interactions are central to lncRNA biology, yet the manuscript primarily focuses on secondary/tertiary probing and RNP methods. Key technologies developed specifically for mapping RNA duplexes—such as KARR-seq, PARIS, RIC-seq, and related psoralen-based crosslinking approaches—are not discussed. These methods are essential for identifying long-range intramolecular interactions and intermolecular lncRNA–RNA contacts. A dedicated subsection summarizing these tools and their contributions to understanding lncRNA structure–function relationships would substantially strengthen the review.
Answer: We thank the reviewer for this thoughtful suggestion. We have added a new subsection (Section 8.1) that specifically addresses the biological relevance of RNA–RNA interactions and provides an overview of key technologies, such as KARR-seq, PARIS, RIC-seq, and other psoralen-based crosslinking approaches, used to map RNA duplexes.
- The authors should also discuss how post-transcriptional RNA modifications (e.g., m6A, m5C, pseudouridine) reshape lncRNA structure and influence lncRNA–protein interactions. These modifications are increasingly recognized as key regulators of lncRNA function.
Answer: We thank the reviewer for this critical point. We have added a new subsection (Section 8.2) that discusses how post-transcriptional RNA modifications modulate lncRNA structure and influence lncRNA–protein interactions. Regarding the detection of RNA modifications, we previously briefly discussed relevant technical aspects in the context of structural probing (Section 5.3.2). To clarify this connection, we have now added the following statement to the end of that section: “Overall, the choice of reverse transcriptase is critical not only for resolving RNA secondary structure but also for enabling the detection of RNA modifications. Here, we highlighted this technical aspect in the context of structural probing, whereas the biological implications of RNA modifications are examined separately in Section 8.2.”

Reviewer 2 Report
Comments and Suggestions for Authors
See attachment please

Author Response
This manuscript by Oh et al. reviews highly novel, important and fast developing field of study with extensive practical implications and could represent a significant value for the readers. However, in its current form it is not easy to understand due to suboptimal organization and sketchy coverage of the material.
Regrettably, the authors have adopted an AI-like writing style, with excessive low information content sentences and general remarks unsupported by references or experimental data. To give just a few examples:
- line 203 ‘Likewise, understanding the structures of nucleic acids broadens our perspectives on various biological processes. As the most iconic example, the double-helix structure of DNA illustrates the efficient flow of genetic information.’
- line 508 ‘Altogether, this newly emerging long-read sequencing technique will broaden our understanding of RNA structural studies.’
- line 625 ‘In recent years, RNA-targeting strategies have garnered increased attention due to advancements in experimental methodologies and computational approaches. The lncRNA structures and interactome are pivotal in these efforts, necessitating a deeper understanding of their roles.’
- line 685 ‘Both lncRNAs and RBPs are known to perform highly diverse and multifunctional roles [166]. This functional versatility suggests that they may not act in a strictly specific manner within particular biological processes, but instead may engage in broad, promiscuous interactions with various molecular partners.’
- line 71 ‘However, several characteristics distinguish lncRNAs from mRNAs. lncRNAs display distinct patterns of expression specific to cell types, developmental stages, and subcellular localization, suggesting intrinsic functional significance.’
Answer: We thank the reviewer for this valuable comment. We have carefully revised all sentences noted in points 1–5 to improve clarity, precision, and scientific rigor. Wherever appropriate, we have replaced vague statements with more concise descriptions supported by relevant literature or specific context. We hope these comprehensive changes address the reviewer’s concerns and significantly strengthen the overall quality of the text.
- Thank you for the helpful suggestion. We have revised the sentence to provide a clearer and more informative explanation.
“Extending these insights to nucleic acids, studies of RNA and DNA structures have proven to be central to understanding biological processes. For example, the double-helical structure of DNA revealed how genetic information is stored, replicated, and transmitted with high fidelity.”
- We agree that the statement was unnecessary and potentially confusing in its original context. Therefore, we have removed it from the manuscript. Thank you for pointing this out.
- Thank you for the comment. The entire section 7 (now it’s section 6), including this part, has been substantially rewritten to improve clarity and conciseness. During the revision, the original sentence was removed, and its essential content was integrated into the newly composed text.
“LncRNAs with well-defined biological functions can serve as therapeutic targets. Therapeutic strategies include directly targeting structured RNA regions or disrupting lncRNA-protein interactions.”
- Thank you for the comment. The entire section 8 (now it’s section 7), including this part, has been substantially rewritten to improve clarity and conciseness. During the revision, the original sentence was removed, and its essential content was integrated into the newly composed text.
“Adaptor proteins represent a subclass of RNA-binding proteins RBPs that link lncRNAs to large protein complexes and broaden the range of possible interactions [182]. Both RBPs and lncRNAs show functional diversity and often exhibit broad interaction profiles rather than strict specificity [183]. Classic examples include the PRC2 chromatin-modifying complex, whose binding to specific lncRNAs is debated [184, 185]. More recent evidence has emerged for adaptor proteins that mediate specific interactions between lncRNAs and chromatin remodeling complexes [39]. Mapping the full set of lncRNA-protein interactions and identifying functionally essential pairs is therefore a key objective.”
- Thank you for the helpful suggestion. We have revised the sentence to provide a clearer and more informative explanation.
“However, several features set lncRNAs apart from mRNAs. Their expression is often tightly restricted to specific tissues, developmental stages, and subcellular compartments, suggesting that lncRNAs play context-dependent functional roles [14, 15]. In addition, lncRNAs are typically expressed at lower copy numbers, consistent with regulatory rather than housekeeping functions.”
- as well as lines 712-720 and sections 7 and 8 almost in their entirety.
Answer: We appreciate the reviewer’s suggestion. We thoroughly rewrote the entirety of the original Sections 7 and 8 (now Sections 6 and 7) to enhance clarity, eliminate redundancy, and ensure that the narrative is better supported by evidence and mechanistic detail. We hope these revisions substantially improve the readability and informativeness of the manuscript.
- To further distract from the novel and important subject of the review the authors attempt to summarize current advances in lncRNA biology, an impossible task given available space. As a result, sections 1-5 and 8 miss important points and are rushed, sketchy and unnecessary, as multiple comprehensive reviews of these areas have been recently published. These sections could be shortened to a one-page introduction. Fig 1 and Table 1 are not relevant to the subject of the review.
Answer: We thank the reviewer for this perspective. Our intention with those sections was to provide essential background for readers who may not be familiar with specific aspects of lncRNA biology that directly contextualize the structural and interactome-focused sections that follow. Because our review addresses a specialized and rapidly evolving field, we considered it important to include concise summaries of concepts that are required to interpret later sections. To address the reviewer’s concerns, we have condensed these sections by removing non-essential details and streamlining the narrative. We also added references to recent, comprehensive reviews to guide readers seeking a deeper background without expanding the manuscript length. We hope these changes orient readers before structure-focused sections and address the reviewer’s concern about scope and length. The following examples illustrate some of the many revisions that have been made.
“For a comprehensive review of RNA definition, see Mattick et al. (2023) [8].”
“For a comprehensive overview of RNA-protein interaction methods, including protein-centric approaches, see Ramanathan et al. (2019) [189].”
- At the same time, multiple items essential for understanding lncRNA structure and interactomes are completely skipped, such as definitions of primary etc. structural levels, common features of lncRNA primary structures, types of lncRNA functional structures, multiple lncRNA conformations and transition rates, stability of functional structures in different environments and methods of determining primary structure. Methods used to study structure and interactomes could be described in more detail.
Answer: We have added definitions for primary structure and descriptions of common features of lncRNA primary structures, as well as methods for their detection. Definitions for secondary and tertiary structures have also been included. We clarified that functional structures of lncRNAs must be determined experimentally. With regard to lncRNA structural dynamics, this represents a large topic beyond the primary scope of this review. Therefore, we have cited seminal studies for readers seeking further information. Methods related to lncRNA structure and interactomes have been substantially revised to improve clarity and understanding. For example, section 8 (now it’s section 7) was completely rewritten.
“The primary structure, defined as the linear nucleotide sequence of lncRNA, can be determined using standard RNA sequencing and can itself have functional significance.”
“Secondary structure refers to local base-pairing interactions within RNA that fold a linear sequence into motifs such as stem-loops, multi-way junctions, and bulges.”
“In contrast, some lncRNAs depend on complex tertiary structures for function. These higher-order three-dimensional architectures involve long-range interactions between distant RNA regions.”
“For a comprehensive review on RNA structural dynamics, see Spitale and Incarnato (2023) [149].”
- Descriptions of methods used to study lncRNA structure and interactomes that are frequently mentioned in the text, are given only in section 6. Frequently term definition is provided after the term was used multiple times (e.g. Cryo-EM is described last but is mentioned in all preceding paragraphs about other methods; ‘PRC2’ defined only on p.18, ‘MS’ on p.20, ‘MD’ as well). This results in repetitive (e.g. lncTCF7 is described in sections 8 and 8.1) and overall disorganized text.
Answer: We appreciate the reviewer’s comment regarding term definitions and repetitive examples. MS and MD are well-established biochemical and computational techniques, and in the context of this review, they appear only as supporting methodologies rather than central concepts. To improve readability, we have added brief clarifying phrases at their first mention (e.g., “mass spectrometry” and “molecular dynamics simulations”), ensuring clarity while keeping the review’s focus on lncRNA biology and structure.
Cryo-EM has been moved to the beginning of the methods section (Section 5.4) to better introduce it before discussing other techniques. For PRC2, which serves as an example of lncRNA interactomes, we streamlined the description to reduce length and potential confusion, cutting approximately half of the original content.
Regarding repetitive examples such as lncTCF7, this reflects a limitation in the field. Detailed mechanistic studies of many lncRNAs are still limited, and most reports focus on phenotypic observations. During revision, for well-characterized lncRNAs that appear repeatedly, we revised the discussion to emphasize different aspects of them relevant to each section. For instance, lncTCF7 was removed from Section 2 and substantially reduced in Section 8 (now Section 7).
Minor comments:
- line 74 ‘Additionally, lncRNAs tend to have lower copy numbers.’ – needs reference
Answer: We thank the reviewer for pointing this out. A relevant reference has been added to support the statement that lncRNAs generally exhibit lower copy numbers.
- line 74 ‘Collectively, these properties confer the molecular specificity underlying lncRNA-mediated regulation’ – not supported by preceding text lines 80-83 – confusing
Answer: We agree with the reviewer that this statement was out of place and potentially confusing. Therefore, it has been removed. We thank the reviewer for highlighting this issue.
- line 118 – did you mean ‘miR-204-5p’?
Answer: Yes, it is miR-204-5p. However, the original reference paper consistently used the notation “miR-204,” and we have retained this usage to respect the source’s use of terminology.
- line 216 – ‘These structural elements are evolutionary conserved in many cases, and have functional significance’ – which elements?
Answer: We have clarified the sentence to reduce ambiguity.
“Because of these physical properties and their length, most lncRNAs can fold into complex secondary and tertiary structures. These secondary structural elements are often evolutionarily conserved and functionally relevant [32, 71].”
- line 230 ‘Besides, the small pool of lncRNAs’ high-resolution structures may contribute to the ongoing debate on whether there are any biological implications of lncRNAs’ structures. Thus, the limited number of high-resolution lncRNA structures should not discourage ongoing efforts to elucidate their structural and functional roles.’ – confusing
Answer: Thank you for the comment. We have revised the text to improve clarity and avoid confusion.
“Rivas et al. (2017) reported that their R-scape method did not detect conserved secondary structure in several lncRNAs, indicating possible sequence-based functions in some lncRNAs [75]. These findings highlight the difficulty of drawing definitive conclusions about the relative contributions of structure versus sequence, in part because high-resolution structural data are available for only a limited number of RNAs. As additional genomes are sequenced and comparative datasets expand, conserved fea-tures—whether structural or sequence-based—may become more readily detectable across lncRNAs. Nonetheless, accumulating evidence supports the conservation of structural elements in lncRNAs”
- line 227 ‘Rivas et al. (2017) claimed that lncRNAs may have no structure at all ‘ - Rivas et al. reported that their computational method, R-scape, does not detect conserved secondary structure in several lncRNAs
Answer: Thank you for pointing this out. We have revised the sentence to more accurately reflect the findings of Rivas et al. “see comment above”
- line 262 ‘independent’ – not clear
Answer: We agree that this term was ambiguous. As it was not essential for conveying the meaning, we have removed it. Thank you for the suggestion.
- lines 274-278 – not appropriate in this location
Answer: Thank you for the helpful comment. We revised this part to ensure that the logical flow between sections is clear. The preceding text discusses how structural features influence lncRNA function, whereas the following section introduces experimental approaches used to study RNA structure. To provide a coherent transition, we clarified that although lncRNA structure is essential for understanding function, only a limited number of lncRNAs have been structurally characterized to date. We then noted that recent methodological advances are gradually expanding this field. The revised passage now reads:
“As discussed above, structural features contribute to lncRNA function. However, only a handful of the ~20,000 annotated human lncRNAs have been investigated using structural approaches. With recent advances in RNA structure determination methods, an increasing number of lncRNAs are now being structurally characterized (Figure 3). The following section discusses these methodological approaches.”
- line 290-292 - not appropriate in this location. What methods are used?
Answer: Thank you for pointing this out. To provide sufficient context and improve readability, we added an overview of commonly used experimental methods:
“To determine lncRNA secondary structure, several experimental approaches are commonly used. Chemical probing methods modify flexible nucleotides, while enzymatic methods cleave structured or unstructured regions. These modifications are typically detected by reverse transcription followed by high-throughput sequencing. When applying these methods, key factors include probe selection, reverse transcriptase selection, and sequencing strategy to accurately capture nucleotide reactivities.”
- line 338 – ‘outperform’ – in what way?
Answer: We replaced the vague term and added specific details.
“Newer SHAPE reagents provide improved signal-to-noise ratios, cellular permeability and performance in cellular environments.”
- line 344 – ‘As such, while reactions between chemical probes and RNA indicate that the RNA is single-stranded in cellulo, it does not necessarily mean that RNA is double-stranded when chemical probes do not react with RNA’ – confusing
Answer: The sentence has been revised to improve precision and readability.
“As such, a lack of reactivity in cellulo does not necessarily indicate a double-stranded region. Non-reactive sites may also result from protection by proteins or other nucleic acids, or from intra- or intermolecular interactions that block probe access [116-118].”
- line 393 ‘Recently, MRT-ModSeq, which combines next-generation sequencing and machine learning, has been developed to detect multiple types of RNA modifications [99]. MRT incorporates mutations during cDNA synthesis in different RT conditions with different divalent cations [98]. Therefore, this novel technique does not need prior chemical treatment before sequencing library preparation’ – confusing and not clarified until line 420
Answer: To improve readability, we separated the explanation into two parts: a concise introduction of MRT-ModSeq, followed by a clear description of how mutations are introduced under altered reverse-transcription conditions. We hope this rephrasing provides a logical, stepwise explanation of how MRT-ModSeq functions.
“Recently, MRT-ModSeq, which combines next-generation sequencing and machine learning, has been developed to detect multiple types of RNA modifications [125].”
“In this method, reverse transcriptases are placed under specific reaction conditions, such as the use of Mn²⁺ rather than Mg²⁺, that reduce their fidelity. Under these low-fidelity conditions, the enzyme misreads chemically modified nucleotides and introduces mutations into the nascent cDNA [120].”
- line 404 ‘High-throughput sequencing allows for the multiplexing of several RNA species with different lengths in a single experiment. This also allows for transcriptome-wide RNA structure analysis.’ – confusing
Answer: We simplified this statement to eliminate redundancy and improve clarity.
“More recently, these approaches have been replaced mainly by high-throughput next-generation sequencing (NGS). By enabling parallel analysis of multiple RNA species in a single experiment, NGS-based methods support transcriptome-scale determination of RNA secondary structure. At present, the combination of chemical probing with high-throughput sequencing is widely used to infer RNA secondary structure. The general principle underlying these approaches is the detection of reverse transcriptase (RT) stops or mutation events, which are mapped back to a reference RNA sequence to generate reactivity profiles
- line 416 ‘These drawbacks are imperative’ – confusing
Answer: The sentence has been revised to explicitly describe the nature of the limitations.
“In these methods, reverse transcriptase prematurely terminates cDNA synthesis upon encountering chemically modified nucleotides, generating truncated cDNA fragments that are subsequently deep sequenced (reviewed in [135]). Interpretation requires distinguishing true modification-induced termination events from natural RNA ends, a challenge that becomes more pronounced for long RNAs due to RNA degradation and incomplete reverse transcription [132].”
- line 458 ‘To neutralize the negative charge and stabilize the structure, RNA requires multivalent cations, such as Mg2+, to these sites’ – not clear
Answer: We substantially revised this section to provide a clearer mechanistic explanation of how multivalent cations stabilize RNA tertiary structures and how lanthanide ions are used experimentally.
“When RNA folds into compact tertiary structures, negatively charged phosphate groups are positioned closely together, creating localized regions of high negative charge [112, 113]. These sites recruit multivalent cations such as Mg2+, which reduce electrostatic repulsion and stabilize the fold [112, 113]. Lanthanide ions, including Tb3+, also bind to these negatively charged regions, but in contrast to Mg2+, Tb3+ promotes RNA cleavage [114]. The resulting cleavage pattern marks local tertiary structures near the bound sites [114]. Combined with deep sequencing, Tb-seq maps these tertiary motifs and enables targeted functional or high-resolution structural analyses [115].].”
- line 470 ‘By doing so, lncRNAs play key roles in regulating cellular functions, as the example of riboswitch shows.’ – confusing
Answer: We clarified the logic by directly linking conformational changes to regulatory outcomes and providing a concrete example. We hope this revision removes ambiguity and strengthens the mechanistic connection.
“Such conformational changes can regulate cellular function. For example, riboswitches alter their structure in response to small molecules, thereby controlling gene expression.”
- line 491 ‘by directly reading long nucleic acid strands and detecting changes in ionic current as molecules pass through nanoscale pores’ – not clear
Answer: We rephrased the sentence to explain the Nanopore sequencing mechanism more directly and clearly. We hope this explanation improves accessibility for readers unfamiliar with the technique.
“by reading individual RNA molecules in full length. As each nucleotide passes through a nanoscale pore, characteristic changes in ionic current allow base identification [154-158].”
- line 503 ‘Moreover, Nanopore sequencing can be applied to study a more diverse array of RNA modifications, in contrast to current techniques that are limited to a few types, such as m6A and 5mC, which are the most common RNA modifications’ – confusing
Answer: We have rephrased this sentence for clarity.
“Moreover, Nanopore sequencing can detect a broader range of RNA modifications than current methods, which are largely limited to only a few of the most common modifications, such as m6A and 5mC [162].”
- line 561 What type of sample was used to study HOTAIR?
Answer: Thank you for the question. We clarified the sample type and expanded the surrounding explanation to improve context.
“For example, in vitro transcribed HOTAIR RNA was examined by AFM, revealing nine structural segments that form well-defined local structures within a flexible global architecture [85]. Dynamic movements of HOTAIR were observed during interactions with genomic DNA, and its flexible nature was further supported by cryo-EM 2D images [85]. AFM can thus complement techniques such as cryo-EM, although it is less commonly used due to its lower resolution.”
- line 583 ‘X-ray crystallography revealed that YTHDC1 recognizes (m6A)UCG RNA like it is a single-stranded conformation, while the stem loop is not completely disrupted’ - confusing
Answer: We revised and expanded the description to clearly explain the structural basis of YTHDC1 recognition and reduce confusion.
“The A-repeat has conserved AUCG tetraloops with m6A modifications [175]. X-ray crystallography indicated that YTHDC1 binds the (m6A)UCG-tetraloop in a manner consistent with single-stranded recognition, while the lower stem-loop remains base-paired [174]. NMR confirmed that YTHDC1 binding locally destabilizes the upper stem, partially loosening the hairpin without full disruption. This partial destabilization further facilitates YTHDC1 binding and modulates the accessibility of additional RNA-binding proteins [174].”
- lines 613-619 - not appropriate in this location
Answer: We revised this portion to improve flow and context. Specifically, we streamlined the preceding discussion and repositioned the emphasis to naturally lead into cryo-EM as an emerging structural tool.
“Other chromatin modifiers and transcription factors also interact with RNA, often without canonical RNA-binding motifs [164, 166, 167]. This highlights the need for structural studies to investigate the detailed regulatory roles of RNA. In this context, cryo-EM has emerged as a leading tool for investigating RNA and RNP complexes due to the recent advances in electron microscopy and computational modeling. Cryo-EM has been applied to large RNP complexes, such as ribosomes, as well as protein-free RNAs, including small riboswitches and portions of SARS-CoV-2 RNA. It also enables analysis of RNAs with dynamic conformations.”
- line 750 ‘Both methods use crosslinking to preserve RNA and protein-binding partners in cellulo covalently’ – confusing
Answer: We have rewritten the entire paragraph containing this expression to enhance clarity and ensure that the argument is presented in a more precise and coherent manner.
“In vivo methods such as RNA antisense purification coupled with mass spectrometry (RAP-MS) and comprehensive identification of RBPs by mass spectrometry (ChIRP-MS) address these limitations [191, 192]. RAP-MS uses long biotin-labeled DNA probes (~120 nt) and UV crosslinking, increasing specificity and detecting only direct interactions [189, 191]. ChIRP-MS uses short probes (~20 nt) and formaldehyde crosslinking, capturing both direct and indirect interactions [189, 192]. Both methods require large cell numbers due to low recovery after crosslinking and hybridization (Table 3) [189].”

Reviewer 3 Report
Comments and Suggestions for Authors
The paper covers an interesting and relevant topic and the effort to address multiple aspects of lncRNA biology is appreciated reflecting a thorough literature search; however, it requires an extensive review to improve clarity and strengthen its overall presentation. The manuscript would benefit from clearer and more direct phrasing, with careful attention to spelling, typos, and formatting. Several references, particularly sections 2.2 and 3, are somewhat outdated and could be updated with more recent literature. Some content is repetitive, for example, HOTAIR and its mechanism are mentioned multiple times across different paragraphs, while many lncRNAs are presented superficially, often focusing on less-studied examples. Citing too many lncRNA examples fragments the text and risks confusing the reader. It may be more effective to highlight well-known lncRNAs, reusing them across sections but emphasizing different features depending on the context. Paragraph 4 could be strengthened, and certain sections (e.g., Paragraph 5: therapeutic applications) feel superfluous and distract from the main focus of the paper. Figures also need clarification: in Figure 3, the table’s reference is unclear, and Figure 4 could be expanded to include enzymatic probes and repositioned to align better with the text. Section 6.3.2 on post-transcriptional RNA modifications appear disconnected from the surrounding discussion and interrupt the flow on secondary structure characterization. Similarly, in section 6.4 the transition from tertiary structure to examples of protein interactors is not clear; it feels somewhat misplaced, especially since a dedicated section on this topic appears later. I would recommend removing this part. Section 8.1 could be developed further by including additional examples, perhaps described more concisely, rather than dedicating the entire section to a single case study. Finally, the conclusion is rather limited. Turning the conclusion into more of a discussion, summarizing the main points, pointing out open questions, and highlighting future perspectives would give the manuscript a better closing.
Author Response
The paper covers an interesting and relevant topic and the effort to address multiple aspects of lncRNA biology is appreciated reflecting a thorough literature search; however, it requires an extensive review to improve clarity and strengthen its overall presentation. The manuscript would benefit from clearer and more direct phrasing, with careful attention to spelling, typos, and formatting. Several references, particularly sections 2.2 and 3, are somewhat outdated and could be updated with more recent literature.
Answer: Thank you for the comment. We have updated several sections, including 2.2 and 3, with more recent references and carefully revised the manuscript for clarity, spelling, and formatting. Here are a few examples:
“For a comprehensive review of RNA definition, see Mattick et al. (2023) [8].”
“More recently, lncRNA SLERT has been shown to regulate transcription by RNA polymerase I by altering the biophysical properties of the fibrillar center and dense fibrillar component in the nucleus [20].”
“For example, mtlncRNA lncMtDloop – named after the region of the mitochondrial genome that encodes the lncRNA – increases mitochondrial regulation and suppresses Alzheimer’s progression [24].”
“Lastly, lncRNAs like HOTAIR and NEAT1 act as scaffolds by binding different regulatory factors and RNA-binding proteins (RBPs) in coordinating gene repression across the genome and driving the formation of stress-induced nuclear bodies [33, 34].”
Some content is repetitive, for example, HOTAIR and its mechanism are mentioned multiple times across different paragraphs, while many lncRNAs are presented superficially, often focusing on less-studied examples. Citing too many lncRNA examples fragments the text and risks confusing the reader. It may be more effective to highlight well-known lncRNAs, reusing them across sections but emphasizing different features depending on the context.
Answer: Thank you for the comment. The manuscript has been extensively revised for conciseness. Sections 7 and 8 (now Sections 6 and 7) were completely rewritten to reduce redundancy. Since detailed mechanistic studies of many lncRNAs are still limited and most reports focus on phenotypic observations, we prioritized removing examples of less-studied lncRNAs during the revision. For well-characterized lncRNAs that appear repeatedly, we revised the discussion to highlight different aspects relevant to each section. For instance, lncTCF7 was removed from Section 2 and substantially reduced in Section 8 (now Section 7).
Paragraph 4 could be strengthened, and certain sections (e.g., Paragraph 5: therapeutic applications) feel superfluous and distract from the main focus of the paper.
Answer: Thank you for the comment. Section 5 has been incorporated as subsection 4.3 of Section 4 and extensively revised for conciseness, reducing the sense that it was out of place. This restructuring also strengthens the focus of Section 4 on the disease relevance of lncRNAs. Additionally, Sections 7 and 8 (now Sections 6 and 7) were entirely revised to improve clarity and conciseness.
Figures also need clarification: in Figure 3, the table’s reference is unclear, and Figure 4 could be expanded to include enzymatic probes and repositioned to align better with the text.
Answer: Thank you for the comment. We have added more detailed references for the table in Figure 3. Regarding Figure 4, we intended to focus on chemical probing methods. Enzymatic probes are now largely outdated and rarely used, whereas new probes continue to be developed for chemical probing. Thus, the figure aims to illustrate the fundamental principles of chemical probing. Finally, the arrangement of figures and tables is determined by the journal editor during the publication process and is not under our control.
“PDB database search was performed for polymer entity types, including protein, DNA, and RNA. All structures were included regardless of refinement resolution.”
Section 6.3.2 on post-transcriptional RNA modifications appear disconnected from the surrounding discussion and interrupt the flow on secondary structure characterization.
Answer: Thank you for the comment. We have revised Section 6.3.2 (Now it’s 5.3.2) to strengthen its connection to the surrounding discussion. The choice of reverse transcriptase is critical for both secondary structure resolution and RNA modification detection. We now explicitly note that this section focuses on the technical aspects of structural probing, while the biological implications of RNA modifications are discussed separately in Section 8.2.
“Overall, the choice of reverse transcriptase is critical not only for resolving RNA secondary structure but also for enabling the detection of RNA modifications. Here, we highlighted this technical aspect in the context of structural probing, whereas the biological implications of RNA modifications are examined separately in Section 8.2.”
Similarly, in section 6.4 the transition from tertiary structure to examples of protein interactors is not clear; it feels somewhat misplaced, especially since a dedicated section on this topic appears later. I would recommend removing this part.
Answer: Thank you for the comment. Some examples included in the descriptions of each technique were overly detailed and potentially distracting. We have therefore streamlined examples that might be confused with protein interactors, reducing the overall length by approximately half compared to the pre-revision version. One example has been revised as follows:
“The PRC2 complex is a chromatin modifier that methylates histone H3 lysine 27 through its enzymatic subunit EZH2 [165]. The 3.3 Å resolution cryo-EM structure of the PRC2-RNA complex revealed the mechanism of RNA-mediated PRC2 inhibition [164]. The G-quadruplex RNA binds and dimerizes PRC2, forming an interface between two EZH2 subunits and blocking nucleosome binding [164]. This illustrates how small RNA domains can mediate interactions with protein partners.”
Section 8.1 could be developed further by including additional examples, perhaps described more concisely, rather than dedicating the entire section to a single case study.
Answer: Thank you for the comment. The section has been revised to be more concise, thereby reducing the impression that it is dedicated to a single case study.
“Many lncRNAs are associated with chromatin-modifying complexes such as SWI/SNF, which regulate both gene activation and repression by altering nucleosome positioning [15, 186, 187]. lncTCF7 recruits the SWI/SNF complex to activate transcription of TCF7, leading to stimulation of WNT signaling and cancer progression [38, 39]. In contrast, UCA1 binds SWI/SNF and inhibits its chromatin association at the p21 promoter, promoting cancer cell proliferation [188].
These examples, among many others, highlight the need to determine how lncRNAs modulate chromatin remodeling. Identifying direct binding partners and defining the corresponding binding sites are essential for understanding lncRNA-mediated gene regulation.”
Finally, the conclusion is rather limited. Turning the conclusion into more of a discussion, summarizing the main points, pointing out open questions, and highlighting future perspectives would give the manuscript a better closing.
Answer: Thank you for the comment. The first two paragraphs of the conclusion now summarize the main points highlighted in the review, while the final paragraph has been revised to suggest future research directions for a deeper understanding of lncRNAs.
“Growing evidence links lncRNA structure to function, highlighting the value of structural studies for uncovering key molecular mechanisms. Current findings indicate that lncRNAs are globally flexible and organized into modular domains that act as protein-binding sites and scaffolds. This intrinsic flexibility complicates structural determination, necessitating integrative approaches that capture dynamic and heterogeneous conformations. Nevertheless, recent methodological advances are increasingly addressing these challenges.
Because lncRNAs rarely act alone, interactome analyses are equally important. Their molecular partners can influence lncRNA folding, serving as structural switches that stabilize or remodel specific regions. We also reviewed emerging tools that expand the scope and resolution of lncRNA interactome mapping. Combined structure-focused and interactome-based analyses will be essential for elucidating the mechanisms of still understudied lncRNAs.
Although structural and interactome studies have substantially advanced the field, RNA modifications remain an underexplored layer that can profoundly affect RNA structure and activity. Exploring how these modifications influence lncRNA architecture and interactions will provide further insight into their diverse functional mechanisms.”

Round 2
Reviewer 1 Report
Comments and Suggestions for Authors
I am satisfied by the revised version.
Reviewer 2 Report
Comments and Suggestions for Authors
Edits have been made in line with my suggestions.
Comments on the Quality of English LanguageLargely acceptable.
Reviewer 3 Report
Comments and Suggestions for Authors
In the revised manuscript, the authors have satisfactorily addressed most of my comments, resulting in improved readability and overall flow.